# RBD-Protein/Peptide Vaccine UB-612 Elicits Mucosal and Fc-Mediated Antibody Responses against SARS-CoV-2 in Cynomolgus Macaques

**DOI:** 10.3390/vaccines12010040

**Published:** 2023-12-29

**Authors:** Shixia Wang, Farshad Guirakhoo, Sivakumar Periasamy, Valorie Ryan, Jonathan Wiggins, Chandru Subramani, Brett Thibodeaux, Jaya Sahni, Michael Hellerstein, Natalia A. Kuzmina, Alexander Bukreyev, Jean-Cosme Dodart, Alexander Rumyantsev

**Affiliations:** 1Vaxxinity, Inc., Merritt Island, FL 32953, USA; fg@expres2ionbio.com (F.G.); valorie@vaxxinity.com (V.R.); jwiggins4827@gmail.com (J.W.); brett@vaxxinity.com (B.T.); jaya@vaxxinity.com (J.S.); michael@vaxxinity.com (M.H.); jc@vaxxinity.com (J.-C.D.); 2Department of Pathology, University of Texas Medical Branch, Galveston, TX 77550, USA; sivavet@gmail.com (S.P.); chsubram@utmb.edu (C.S.); nakuzmin@utmb.edu (N.A.K.); abukreye@utmb.edu (A.B.); 3Galveston National Laboratory, Galveston, TX 77550, USA; 4Department of Microbiology & Immunology, University of Texas Medical Branch, Galveston, TX 77550, USA

**Keywords:** SARS-CoV-2, COVID-19, RBD, subunit, vaccine, antibody, Fc-mediated effector function, ADCP, ADMP, ADNKA, non-human primates

## Abstract

Antibodies provide critical protective immunity against COVID-19, and the Fc-mediated effector functions and mucosal antibodies also contribute to the protection. To expand the characterization of humoral immunity stimulated by subunit protein–peptide COVID-19 vaccine UB-612, preclinical studies in non-human primates were undertaken to investigate mucosal secretion and the effector functionality of vaccine-induced antibodies in antibody-dependent monocyte phagocytosis (ADMP) and antibody-dependent NK cell activation (ADNKA) assays. In cynomolgus macaques, UB-612 induced potent serum-neutralizing, RBD-specific IgG binding, ACE2 binding-inhibition antibodies, and antibodies with Fc-mediated effector functions in ADMP and ADNKA assays. Additionally, immunized animals developed mucosal antibodies in bronchoalveolar lavage fluids (BAL). The level of mucosal or serum ADMP and ADNKA antibodies was found to be UB-612 dose-dependent. Our results highlight that the novel subunit UB-612 vaccine is a potent B-cell immunogen inducing polyfunctional antibody responses contributing to anti-viral immunity and vaccine efficacy.

## 1. Introduction

The pandemic of the coronavirus disease 2019 (COVID-19) has resulted in over 770 million confirmed cases, including 6.9 million cumulative deaths globally and over 13 billion vaccine doses administered by September 2023 [1]. All COVID-19 vaccines incorporate the spike glycoprotein (S) of SARS-CoV-2 or its receptor-binding domain (RBD) as the key immunogens to elicit neutralizing antibodies crucial for protection [2,3,4,5]. In addition to neutralizing antibodies, growing evidence demonstrated the importance of the antibody Fc-dependent effector functions in protecting from SARS-CoV-2 infection [6].

Non-neutralizing antibodies and Fc-mediated antibody effector functions have been reported as an essential part of the humoral immune responses against viral infections, such as RSV [7], Ebola virus [8], HIV [9], and SARS [10]. The Fc-mediated effector functions serve as a link between the adaptive and innate immune responses and include antibody-dependent cell-mediated cytotoxicity (ADCC), antibody-dependent cellular phagocytosis (ADCP), antibody-dependent monocyte phagocytosis (ADMP), and antibody-dependent nature killer (NK) cell activation (ADNKA).

The SARS-CoV-2 S-specific therapeutic monoclonal antibodies (mAbs) and the Convalescent Plasma (CONCOR-1) with high levels of neutralization activity and Fc-dependent effector functionality (such as ADCC) achieved a significant reduction in severe outcomes and death [11,12]. The Fc-mediated antibody effector functions induced by mRNA vaccines demonstrated protective immunity against COVID-19 [13], and the Fc-γR-dependent antibody effector functions were required for vaccine-mediated protection against antigen-shifted variants of SARS-CoV-2 [14]. Additionally, mucosal antibody responses in the respiratory tract could reduce the risk of COVID-19 infection [15]. The ability of newly emerging Omicron variants to escape neutralization highlights the potential importance of eliciting neutralizing, Fc-mediated antibody responses and mucosal antibodies for next-generation vaccines.

UB-612 is a novel protein/peptide subunit vaccine, which contains a Chinese hamster ovary (CHO)-expressed RBD fused with a single-chain Fc protein (S1-RBD-sFc, RBD-sFc) and five peptides from the N, M, and S2 proteins of SARS-CoV-2 s (Figure 1).

The composition of UB-612 is unique among COVID-19 vaccines; it provides the advantage of stimulating high levels of neutralizing antibodies as well as a Th/CTL response in the vaccinated host, which mimics the immune response that occurs after natural infection with SARS-CoV-2. Furthermore, the breadth of antigen components in UB-612 is anticipated to help recall the pre-existing immunity after COVID-19 vaccination or natural infection [16,17,18]. Both preclinical and clinical studies demonstrated that UB-612 induced robust RBD- and S-specific antibodies with broad neutralizing activity against prototype SARS-CoV-2 viral strains and multiple viruses of concern or interest, including the Alpha, Beta, Gamma, Delta, and Omicron variants. The preclinical studies demonstrated that UB-612 elicited protective immunity against SARS-CoV-2 intranasal and intratracheal challenges in human angiotensin-converting enzyme 2 (hACE2)-transduced mice, rhesus macaques, and cynomolgus macaques [16,17,18]. In the cynomolgus macaques, two-dose immunization with UB-612 induced high titers of RBD-specific and -neutralizing antibodies and reduced the viral loads in the upper and lower respiratory tracts after the SARS-CoV-2 challenge [19]. The current study characterized non-neutralizing antibody responses after UB-612 vaccination in cynomolgus macaques. The new results demonstrate that UB-612 elicited high-avidity RBD-specific antibodies with predominant IgG1 and IgA subclasses and Fc-mediated ADMP and ADNKA antibody responses against SARS-CoV-2.

## 2. Materials and Methods

UB-612 vaccine formulation. As previously reported, the UB-612 vaccine immunogen formulation used for the macaque immunization study comprised RBD-sFc protein and five conserved Th/CTL S2/M/N peptides [19]. RBD-sFc protein was produced from a stable CHO cell line. Additional components of the formulation comprised CpG1, Adju-Phos, and inert salt. CpG1 is a 32-mer type B oligodeoxynucleotide sequence capable of adjuvant activity but used here as an excipient at a dose too low to elicit measurable adjuvant activity (4 μg/mL, which is well below 0.5–3 mg per dose used in various vaccines) [20,21,22]. As documented in previous publications and patents, CpG binds the positively charged peptides by electrostatic interactions, stabilizing the peptides and providing enhanced immune responses at low doses of both CpG and peptide [23,24,25]. Adju-Phos^®^ (Invivogen, San Diego, CA, USA) was used as an adjuvant. In UB-612 drug product formulation, the weight ratio of immunogens is 88% for the RBD-sFc protein and 12% for T-cell peptides.

Cynomolgus macaque immunization. Three groups (*n* = 5 per group) of cynomolgus macaques, 3–6 years old, were immunized with UB-612 formulations containing 30 μg or 100 μg antigens or saline on Days 0 and 28 at Biomere in a facility accredited by the Association for Assessment and Accreditation of Laboratory Animal Care (AAALAC). The study was performed according to Institutional Animal Care and Use Committee (IACUC) guidelines. The injection was administered intramuscularly to the left deltoid muscle; the right side was selected only if the left side was found unsuitable. Serum samples were collected on Days 0, 14, 28, and 50 to evaluate RBD-specific antibody responses as previously described [19]. The serum samples collected three weeks after the second immunization (Day 50) were used for the RBD-specific antibody and Fc-mediated function evaluations. The bronchoalveolar lavage fluids (BAL) collected on Day 50 were used for mucosal antibody evaluation.

Enzyme-linked immunosorbent assay (ELISA). Four types of ELISAs were performed to evaluate the RBD-specific antibody titers in serum and BAL mucosal samples. The IgG titers and IgG isotype titer results were expressed as the reciprocal of the highest serum dilution with a reading of two standard deviations over the cut-off, negative control background.

RBD-specific IgG titers in serum and BAL samples: The serum (baseline and three weeks after the second immunization) and BAL mucosal samples (three weeks after the second immunization) were analyzed by ELISA using the ACRO Biosystems (Newark, NJ, USA) Monkey Anti-SARS-CoV-2 Antibody IgG Titer Serological Assay Kit (RAS-T019) with proper serum and BAL sample dilutions as previous described [19].

IgA antibody in BAL samples: ELISA was conducted using the ACRO Biosystems Monkey Anti-SARS-CoV-2 Antibody IgG Titer Serological Assay Kit (RAS-T019) as above but with HRP-conjugated anti-monkey IgA to replace the secondary anti-monkey IgG antibody in the kit.

RBD-specific antibody avidity: Serum samples were subjected to IgG avidity ELISA using potassium thiocyanate as a chaotropic chemical reagent as previously described [26] to measure the IgG binding strength to RBD. ELISAs were performed as described above for the IgG titer measurement, except an extra step was performed after the incubation with serum. Briefly, after serum incubation and washes, wells were treated with KSCN 1.5 M (200 µL/well) for 20 min at room temperature. After washes, the plates were further incubated with goat anti-monkey IgG HRP-antibody for 1 h at room temperature. Following One Step-TMB substrate incubation, the reaction was stopped by 2N H_2_SO_4_, and absorbance was read with a microplate reader (Spectramax, Molecular Device, San Jose, CA, USA). Avidity Index (AI%) was expressed as follows: AI% = (OD mean value from KSCN-treated sample divided by the OD mean value from the non-treated one) multiplied by 100. The AI values above 50% were considered high antibody avidity; between 31% and 49%, intermediate avidity; and below 30%, low avidity.

RBD-specific IgG subclass isotyping: S1-RBD-specific antibody titers for NHP IgG subtypes (IgG1, IgG2, IgG3, and IgG4) were measured by ELISA. Thermo Fisher Maxisorp 96-well plates were coated with 1 µg/mL of recombinant SARS-CoV-2 spike protein (aa319-541, Thermo Fisher Scientific, Waltham, MA, USA), diluted in phosphate-buffered saline (PBS, 100 µL/well), and incubated for 1 h at 25 °C followed by overnight incubation (16–18 h) at 4 °C. The plates were washed and treated with a blocking reagent before adding NHP sera. Serially diluted NHP serum samples (3-fold diluted from 1:100 to 1:218,700, total eight dilutions) in replicates of 2 were then applied to plates (100 μL/well) before incubating at 37 °C for 1.5 h on an orbital shaker (300 rpm). Rabbit anti-baboon IgG1, IgG2, IgG3, and IgG4 antibodies (Absolute Antibody, Wilton, UK) were diluted 1:5000 in antibody diluent (2% BSA in PBS), applied to plates (100 μL/well), and then incubated at 37 °C for 1.25 h on an orbital shaker (300 rpm). Mouse anti-rabbit IgG linked to horseradish peroxidase (Invitrogen) diluted 1:5000 in antibody diluent was added to each well (100 μL/well), followed by the addition of 3,3′,5,5′-tetramethylbenzidine (1 Step Ultra TMB-ELISA substrate, Thermo Scientific, Waltham, MA, USA). Finally, 1.0 M sulfuric acid was added to each well to terminate the reaction, and the absorbance of each well was measured within 10 min at 450 nm using a microplate reader (Spectramax).

Human ACE-2 binding-inhibition assay. The RBD:ACE-2 binding-inhibition assay was performed by a plate-based SARS-CoV-2 surrogate virus-neutralizing test (sVNT) (GenScript, Piscataway, NJ, USA). In a method similar to those described [27], serum samples were mixed with HRP-conjugated RBD protein and incubated for 30 min. Mixtures were then added to wells coated with ACE-2 and incubated for 15 min. Unbound RBD-HRP was washed from the wells, and reactions were developed with TMB. Absorbance (OD 450 nm) was measured in each well. The percent (%) sRBD-HRP:ACE-2 binding inhibition was calculated using the formula (1 − (sample OD/negative control OD)) × 100. The 50% inhibitory dose (ID50) titers were calculated.

Neutralization assessment by plaque reduction assay. Serum samples were heat-inactivated for 30 min at 56 °C and underwent 2-fold serial dilution in 96-well U-bottom plates in MEM (Gibco, Waltham, MA, USA) containing 2% FBS and 0.1% gentamicin sulfate. A total of 50 µL of each diluted serum sample was mixed with 100 plaque-forming units (PFUs) of virus in 50 µL. The serum/virus mixtures were incubated for 1 h at 37 °C. A total of 50 µL of the serum/virus mixtures were then transferred to Vero E6 cell monolayers in flat-bottom 96-well plates and incubated for 1 h at 37 °C. The serum/virus mixture was then removed and replaced with a 1:1 overlay composed of 1.1% methylcellulose (Fisher Chemical, Waltham, MA, USA) and 2X MEM (Gibco) supplemented with gentamicin sulfate and 4% FBS (Gibco). Plates were incubated for two days at 37 °C, then fixed with 10% neutral-buffered formalin (NBF). Viral plaques were counted after staining for 30 min with 1% crystal violet in formalin (100 µL per well) at room temperature, followed by 5–7 rinses with water. In all neutralization assays, IC50 values were calculated using GraphPad Prism 9.3.1 software. The detection limit was dilution 1:10 for the PRNT50 assay.

Fc-medicated effector antibody functions. The Fc-mediated antibody effector assays were performed as described previously [28].

Antibody-dependent monocyte phagocytosis (ADMP) assay. Vero E6 cells (ATCC, Manassas, VA, USA) were infected with the SARS-CoV-2-WA-GFP virus generated in Dr. Bukreyev’s laboratory at a multiplicity of infection (MOI) of 0.1 and were cultured at 37 °C. After 48 h, the virus-infected Vero cells were collected and incubated with serum (1:20 dilution) for 90 min. Monocytes were isolated from human PBMCs and co-cultured with the virus-infected Vero cells for 4 h. The monocytes were surface-stained with anti-CD14 and anti-CD66b antibodies for 10 min, and cells were washed and fixed with 4% paraformaldehyde (PFA) overnight. The live virus assay steps were performed in a biosafety level 3 (BSL-3) facility at UTMB. Cells were analyzed in an LSR Fortessa analyzer (BD Biosciences, Franklin Lakes, NJ, USA) with FlowJo software version 10.8 (BD Biosciences, Franklin Lakes, NJ, USA). The CD14^+^ cells loaded with virus-infected GFP^+^ cells were quantified to calculate the percent change in phagocytosis of any antibody sample over the control without the antibody added. Serum greater than a 1.8-fold increase over the No-Serum control was categorized as positive for ADMP.

Antibody-dependent NK cell activation (ADNKA) assay. Target cells (SARS-CoV-2-infected Vero cells) were harvested (as performed for the ADMP assay) and preincubated at 37 °C with 1:20 diluted serum for 90 min. Human Peripheral Blood Mononuclear Cells (PBMCs, Gulf Coast Regional Blood Center, Houston, TX, USA), serving as effector cells isolated from the fresh blood, were added to the Vero cell–serum mixture at a 10:1 ratio (effector:target). The cells were incubated at 37 °C for 5 h in the presence of GolgiStop (0.7 μL/mL, BD Biosciences), Brefeldin-A (1:1000, BioLegend, San Diego, CA, USA), and anti-CD107a–FITC (clone H4A3, BioLegend) antibody. After co-culture, cells were stained with live/dead Fixable Aqua (Thermo Fisher Scientific), anti-CD3–APC/Fire750 (clone SK7, BioLegend), and anti-CD56–BV605 (clone 5.1H11, BioLegend) at room temperature for 10 min. Cells were washed once with PBS and fixed with 4% PFA twice per BSL-3 laboratory protocol. The assay steps using live virus were performed in a BSL3 facility. The fixed cells were moved to a BSL-2 laboratory, where intracellular cytokine staining (ICS) was performed. For ICS, cells were incubated with permeabilization buffer (BioLegend) for 20 min and stained with an antibody cocktail containing anti-IFN-γ-APC (Clone 4S.B3, BioLegend) and anti-tumor necrosis factor (TNF) a-BV421 (Clone Mab11, BioLegend) for 30 min. Cells were washed once with PBS and were analyzed in an LSR Fortessa analyzer with FlowJo software version 10.8 BD Biosciences. NK cells were identified as CD3^−^CD56^+^ cells from the lymphocyte pool. The ADNKA cell activation was quantified as the proportion (%) of NK cells with activation markers CD56^+^CD107^+^, CD56^+^ IFN-γ^+^, or CD56^+^TNFα^+^. A ≥2-fold change in percent of activated NK cells for any serum relative to no serum was considered optimal activation of NK cells.

Statistical analyses. All statistical analyses were performed using GraphPad Prism 9.2.0. The data presented in the graphs are group geometric mean titers/values (GMTs) with standard deviation (SD). An unpaired *t*-test or one-way analysis of variance (ANOVA) was performed to compare variables between different groups. *p*-values ≤ 0.05 were considered as significant differences.

## 3. Results

### 3.1. Subsection RBD-Specific Binding Antibody Responses

The cynomolgus macaques received 30 μg or 100 μg of UB-612 or saline on Days 0 and 28. After the first dose, both dose groups elicited RBD-specific IgGs, which were further boosted by the second immunization with the high titers of RBD-specific antibody responses detected in both 30 μg (mean titer 1:47,104) and 100 μg (mean titer 1:46,771) dose groups. As expected, no RBD-specific antibody responses were detected in the saline group (Figure 2A). The RBD-specific IgG had high binding avidities, with a mean Avidity Index (AI) of 73.5% in the 30 μg dose group and 68.8% in the 100 μg dose group (Figure 2B).

To profile the RBD-specific IgG isotypes in immune macaques, we performed IgG isotyping ELISA. In the 30 μg dose group, the mean endpoint IgG1 titer was 1:17,820, while the IgG2, IgG3, or IgG4 titers were very low (1:160, 1:150, and 1:120). Similarly, in the 100 μg dose group, the mean endpoint IgG1 titer was 1:11,340, while the IgG2, IgG3, or IgG4 titers were very low (1:220, 1:60, and 1:50) with IgG1:IgG2, IgG3, or IgG4 ratios over 50-fold (Figure 2C).

### 3.2. ACE-2 Binding-Inhibition and Neutralizing Antibody Responses

Although the RBD-specific binding antibody titers above were similar between the 30 μg and 100 μg doses, the ACE-2 binding-inhibition and neutralizing antibody activities showed dose-dependent responses. The ACE-2 binding-inhibition antibody assessments showed that the 100 μg dose had mean ID50 titers of 2527.2, which is 1.6-fold higher than the 30 μg dose (ID50 titers of 1588.2, *p* < 0.0001) (Figure 3A). In the present study, the neutralizing antibody responses are reported only against the SARS-CoV-2 WA strain. The mean NAb titer in the 100 μg dose group was 1:1100 higher than the 30 μg dose group (mean NAb titer 1:715). However, the difference was not statistically significant (*p* = 0.46) due to each group’s wide range of titer variations (Figure 3B).

### 3.3. Mucosal IgG and IgA Antibodies in BAL

In the previously published findings, we demonstrated that the 2-dose UB-612 vaccination protected both the lower and upper airways by significantly reducing viral subgenomic RNA (sgRNA) loads in BAL and nasal swabs compared to the saline group [19]. Individual macaque BAL samples collected three weeks after the second immunization were analyzed for the RBD-specific IgG and IgA antibodies to evaluate the mucosal antibody responses in the airway. BAL IgG and IgA responses were detected in the vaccine group but not in the saline group. The mean BAL IgG titers were 1:5.5 in the 30 μg dose group and 1:10.4 in the 100 μg dose group, while the mean titer in the saline group was 1:0.3, which was not significantly lower compared to the vaccine groups (Figure 4A). The mean BAL IgA titers were 1:2.4 in the 30 μg dose group and 1:3.1 in the 100 μg dose group, while the mean titer in the saline group was 1:1.2. Only two BAL samples were available for the IgA measurement; therefore, it was not possible to determine the significance in the difference of IgA responses between the treatment groups.

### 3.4. Antibody-Dependent Monocyte Phagocytosis Responses

We conducted ADMP assays to detect human monocyte phagocytosis activity of UB-612 immune macaque sera against SARS-CoV-2-WA-GFP-infected Vero E6 cells. After the co-culture of infected Vero E6 cells and monocytes, CD66b^+^/CD14^+^ monocytes were analyzed for the GFP-positive signal: the RBD-specific antibodies bound to the infected Vero cells engaged the monocytes via the FcyR to the phagocyte-infected cells expressing GFP, (Appendix A). Compared with the saline-immunized macaques, the UB-612-immunized macaque sera promoted more significant ADMP activity against the WA strain in a vaccine dose-dependent trend (Figure 5A). The mean ADMP activity in the 30 µg dose group was 6.7%, which was 1.9-fold higher than the saline group (*p* < 0.05) and 2.6-fold higher than the baseline bleeds (*p* < 0.01). The mean ADMP activity in the 100 µg dose group was 7.6%, which was 2.2-fold higher than the saline group (*p* < 0.001) and 3-fold higher than the baseline (*p* < 0.0001).

### 3.5. Antibody-Dependent NK Cell Activation Responses

To further assess the Fc-mediated effector functions induced by UB-612 immune macaque sera, we performed ADNKA assays to detect NK cell activation against SARS-CoV-2-infected Vero E6 cells. After the co-culture of infected Vero E6 target cells with human PBMCs as effector cells, CD3^−^/CD56^+^ NK cells were analyzed for CD107a degranulation and IFN-γ or TNF-α production as activation signals (Appendix A). Compared to the saline-immunized macaques, UB-612-immunized macaque sera promoted stronger ADNKA activities against the WA strain in a dose-dependent manner, with a higher proportion of CD107a- and IFN-γ-positive CD56 NK cells detected between the 100 µg UB-612 group and saline, but no difference between the treatment groups was observed for TNF-α-positive CD56 NK cells (Figure 5B–D).

The mean proportion of CD107a degranulation in the 30 µg dose group was 29.7% of CD56 cells, which is 3.3-fold higher than the saline group (*p* < 0.001) and 1.4-fold higher than the baseline (*p* < 0.01); the mean CD107a cell proportion in the 100 µg dose group was 54.3%, which is 6.2-fold higher than the saline group (*p* < 0.0001) and 2.5-fold higher than the baseline (*p* < 0.001); and the CD107a cell proportion in the 100 µg dose group is significantly higher than the 30 µg dose group (*p* < 0.0001), as shown in Figure 5B.

The mean IFN-γ production in the 30 µg dose group was 0.25% of CD56 cells, which is 3.1-fold higher than the saline group and 6.5-fold higher than the baseline (*p* < 0.05); the mean IFN-γ production in the 100 µg dose group was 0.35%, which is 4.2-fold higher than the saline group (*p* < 0.001) and 8.4-fold higher than the baseline (*p* < 0.001); and the IFN-γ production in the 100 µg dose group was higher than the 30 µg dose group although this was not statistically significant (*p* = 0.57) due to the higher sample variability, as shown in Figure 5C.

There was no TNF-α production in the 30 µg dose group, similar to the saline group and baseline values. Low TNF-α production was detected in the 100 µg dose group; it was higher compared to the baseline (*p* < 0.001), but it was not different from the saline group (*p* = 0.24), as shown in Figure 5D.

## 4. Discussion

Although neutralizing antibodies are a well-established essential correlate of vaccine-mediated protection from SARS-CoV-2 infection [29], the protection against severe COVID-19 was typically maintained despite waned neutralizing antibody titers and the emergence of SARS-CoV variants [30,31]. Additional protective immune mechanisms against SARS-CoV-2 infection and COVID-19 should also be considered, such as T-cell responses, non-neutralizing antibody responses, and antibody Fc-mediated effector functions [14,32,33,34,35,36,37]. Recent studies have shown that the antibodies induced by spike-protein-based vaccines could be associated with multiple Fc effector functions, including ADCC and ADCP [6,14]. Vaccination and passive antibody transfer to animals demonstrated that the antibody-mediated activation of cellular immunity is a critical component of the protective immune response [38,39,40]. The Fc functional antibodies, including ADCD and ADNP, correlate with protection in rhesus macaques [41]. In humans, the activity of antibody Fc effector functions were retained against antigenically distant SARS-CoV-2 strains [11,12].

UB-612, Vaxxinity’s COVID-19 investigational vaccine, induced robust and broad RBD-binding and -neutralizing antibodies, as well as peptide-specific T-cell responses against the prototype and emerged SARS-CoV-2 strains that protected against SARS-CoV-2 infection in preclinical studies [16,17,18,19]. In the previous study, the protection against SARS-CoV-2 replication in the respiratory tract of NHP directly correlated with the dose of UB-612 [16,17,18,19]. The observed difference in the serum-neutralizing antibodies between the NHP treatment groups was minimal; however, there was evidence of dose-dependent T-cell responses. The new study investigated other immune response factors that could have contributed to the UB-612-induced protection. In the current study, we evaluated the activity of RBD-specific antibodies, including antibody-binding avidity, IgG isotypes, mucosal antibodies, and Fc-mediated effector functions in cynomolgus macaques, a well-accepted animal model for SARS-CoV-2 infection [42]. Active immunization of NHP with UB-612 stimulated antibodies with ADMP and ADNKA activities against SARS-CoV-2 in vitro. The ADMP- and ADNKA-mediated phagocytosis and killing of infected cells were promoted through antibody linking of viral antigens presented on the surface of infected cells with the monocyte and NK cell Fc receptors, respectively, to promote the phagocytosis and killing of infected cells [43]. NK cells are essential in bridging the innate and adaptive responses, and individuals with NK cell deficiencies suffer from severe viral infections [44]. Other COVID-19 vaccines also showed to induce antibodies active in ADCP and NK degranulation against several SARS-CoV-2 variants [45]. Clinical experience indicated that the Fc-dependent antibody effector functions and T-cell responses are essential in vaccine-mediated protection, especially when neutralizing antibody titers are low [13,46,47].

The Fc effector functions are typically associated with IgG1 and IgG3 antibody subclasses [48]. Our analyses demonstrated that IgG1 was the predominant subclass of RBD-specific antibodies induced by the UB-612 vaccine in macaques and likely the one correlated with the ADMP and ADNKA functions detected in the vaccine immune sera.

Antibody avidity defines the binding strength between an antibody and its specific epitope and is also a critical factor in describing antibody maturation [49]. The significance of high-avidity antibodies in protective immunity against SARS-CoV-2 has been demonstrated in various studies [50,51,52,53]. The UB-612 vaccine induced high-avidity RBD-specific antibody responses in macaques, and we believe a substantial fraction of these high-avidity antibodies represent virus-neutralizing antibodies that contributed to the breadth of neutralization against multiple variants of concern observed previously [16,17,18,19].

In addition to serum antibody responses, the mucosal antibodies are critical as the first line of defense in clearing the virus and blocking the viral entry at the mucosal surfaces [54,55]. Studies have shown that mucosal antibody responses contributed to the protective immunity against COVID-19 [56,57,58]. UB-612 elicited both IgG and IgA in the respiratory tracts in macaques, and the BAL-secreted antibodies likely contributed to the protection against SARS-CoV-2 intranasal and intratracheal challenges observed in our preclinical studies [19].

In summary, UB-612 elicited robust multifunctional antibody responses against SARS-CoV-2, including RBD-specific-binding IgG, virus-neutralizing antibodies, Fc-mediated effector functions, and mucosal antibodies. The importance of multifunctional antibody responses has been demonstrated in contributing to protection in multiple preclinical and clinical COVID-19 vaccine studies. The new insights into the broad functionality of vaccine-induced humoral immunity and previously characterized T-cell responses against non-spike antigens provided additional preclinical evidence of UB-612 protective potential and differentiation. The UB-612 vaccine has been proven safe and highly immunogenic in Phases 1 and 2 clinical studies. It is currently being tested as a booster vaccine in a Phase 3 study with promising results supportive of UB-612 non-inferiority or superiority in stimulating neutralizing antibodies compared to several well-established COVID-19 vaccine platforms [16,17,18].

## Figures and Tables

**Figure 1 vaccines-12-00040-f001:**
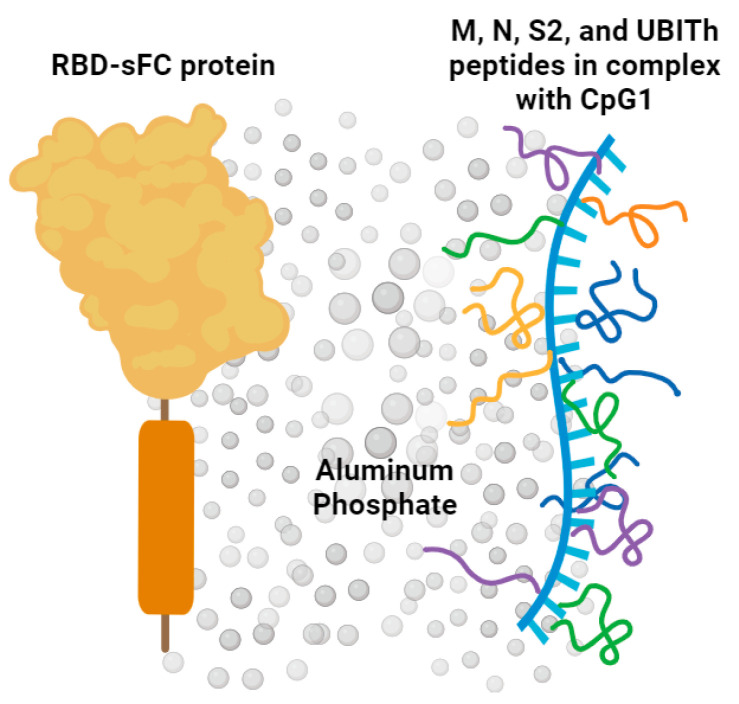
Composition of subunit protein–peptide COVID-19 vaccine UB-612. Receptor-binding domain (RBD) of SARS-CoV-2 spike protein fused to a modified single-chain human IgG1 Fc. Five synthetic peptides incorporating conserved helper and cytotoxic T lymphocyte (Th/CTL) epitopes derived from SARS-CoV-2 structural proteins: three from the S2 subunit, one from membrane, and one from nucleocapsid, and one universal Th peptide. Peptides are precipitated by charge interactions with CpG1 ODN to form a stable complex. The protein and peptides are adsorbed on aluminum phosphate adjuvant.

**Figure 2 vaccines-12-00040-f002:**
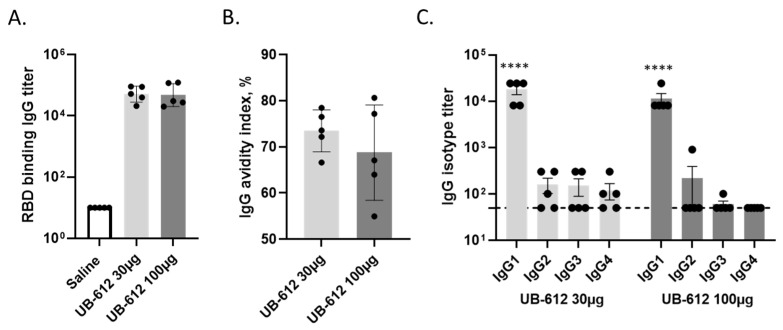
RBD-specific binding antibody responses in macaques three weeks after a 2-dose UB-612 immunization: RBD-binding IgG titers (**A**); Avidity of IgG binding to RBD, expressed as Avidity Index (AI% = OD mean value ratio between KSCN-treated and -non-treated sera) (**B**); and IgG isotypes (**C**). IgG titers and IgG isotypes data are expressed as the highest serum dilution with a reading of two standard deviations over the background cut-off). Presented are the individual value group means with standard deviations. Statistical significance is presented as **** *p* < 0.0001 to compare IgG1 with IgG2, IgG3, or IgG3.

**Figure 3 vaccines-12-00040-f003:**
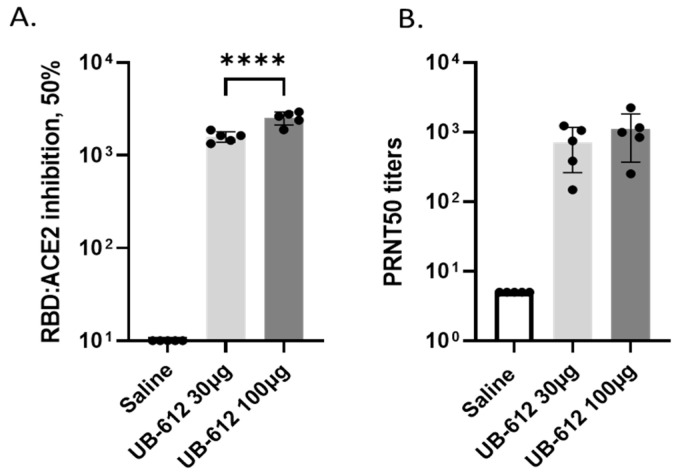
Functional active anti-viral antibody responses three weeks after the 2-dose UB-612 vaccinations. (**A**) RBD:ACE2 binding-inhibition activities (serum dilution resulting in 50% binding inhibition). (**B**) Neutralizing antibody titers against SARS-CoV-2-WA (serum dilution resulting in 50% plaque reduction). Presented are the individual values and group means with standard deviations. Statistical significance between conditions determined by one-way ANOVA is presented as **** *p* < 0.0001.

**Figure 4 vaccines-12-00040-f004:**
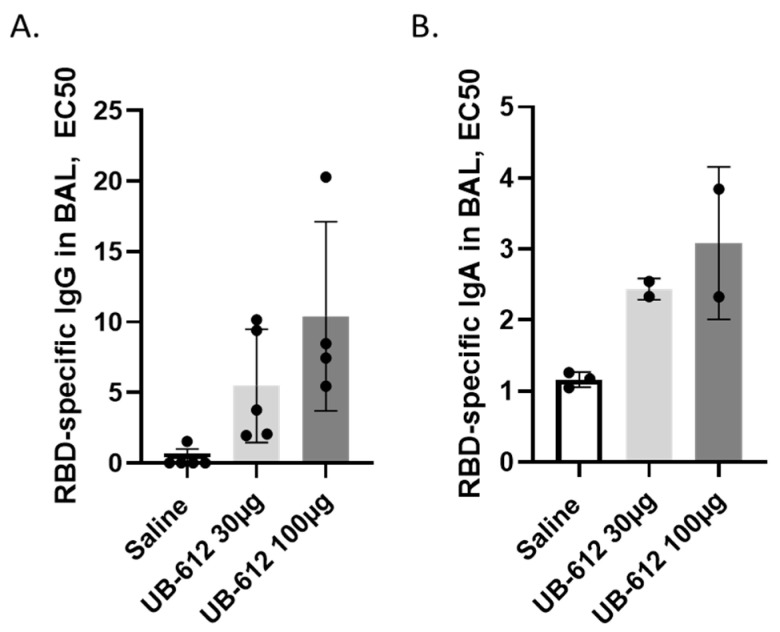
Mucosal IgG (**A**) and IgA (**B**) titers (dilution resulting in EC50) in macaque bronchoalveolar lavage fluids (BAL) were collected three weeks after the 2-dose UB-612 vaccinations. Presented are the individual values and group mean with standard deviations. The IgA was measured only in n = 3 (saline) and n = 2 (UB-612) immunized NHPs.

**Figure 5 vaccines-12-00040-f005:**
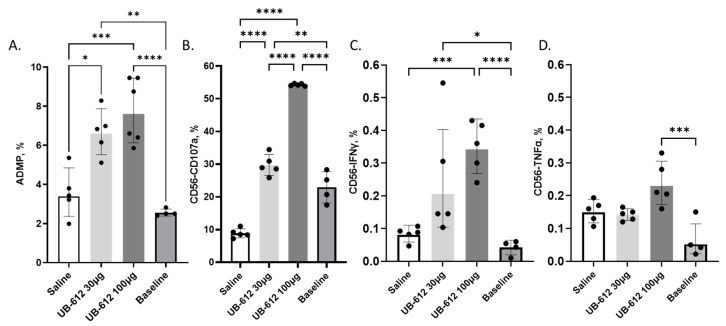
Antibody-dependent monocyte phagocytosis (ADMP) and antibody-dependent NK cell activation (ADNKA) in UB-612-immunized macaques. Sera were collected three weeks after the second immunization. (**A**) ADMP: change in the proportion (%) of GFP-positive monocytes due to ADMP activity in the presence of vaccine immune sera compared to the negative controls with no sera added. ADNKA (**B**–**D**) (**B**) Proportion (%) of CD107a-positive CD56 NK cells. (**C**) % IFN-γ-positive CD56 NK cells. (**D**) % TNF-α-positive CD56 NK cells. Presented are the individual values and group means with standard deviations. Statistical significance between conditions by using one-way ANOVA is presented as * *p* < 0.05, ** *p* < 0.01, *** *p* < 0.001, and **** *p* < 0.0001.

## Data Availability

All reasonable requests for data associated with this manuscript should be routed to the corresponding author.

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
