# Peer review of "RBD-Protein/Peptide Vaccine UB-612 Elicits Mucosal and Fc-Mediated Antibody Responses against SARS-CoV-2 in Cynomolgus Macaques"

_vaccines, 2023, doi:10.3390/vaccines12010040_

Round 1

Reviewer 1 Report

Comments and Suggestions for Authors

In this manuscript, the authors report the characterization of the mucosal and Fc-mediated antibody responses elicited with RBD-protein/peptide vaccine UB-612 against SARS-CoV-2 in cynomolgus macaques. And in addition to the potent serum neutralizing and ACE2 binding inhibition antibodies, the mucosal antibodies and antibodies with Fc-mediated effector functions in antibody-dependent monocyte phagocytosis and antibody-dependent NK cell activation also contribute to the protection. Although the similar studies relevant to other types of vaccines have been reported, the results in this manuscript may prove that UB-612 is a promising COVID-19 vaccine that may provide a protection for the breath variants. The results overall are interesting.

The following suggestions are majorly intended to improve the information accuracy of the manuscript written and result interpretation.

1.    In the Abstract, the authors did not clarify the question or study goal that this study targeted to solve, in addition to lacking the method information. 

2.    Lacking immunization method (inject route) information in the section of Materials and Methods for the immunization of animal cynomolgus macaques.

3.    In the section of Results.

In Figure 2A, by eye measuring, the RBD binding IgG titer (EC50) detected in UB-612 30μg dose group is little bit higher than that detected in UB-612 100μg dose group. While in the text line 215, the RBD-specific antibody response (mean titer 1:47,544) detected in UB-612 30μg dose group is little bit lower than that detected in UB-612 100μg dose group (mean titer 1:51,333).

In lines 237 – 239, the ACE-2 binding inhibition antibody assessments showed that the 100μg dose group (mean ID50 titers of 2,527.2), although it was statistically higher than the 30μg dose group (ID50 titers of 1,588.2 < 0.0001) in Figure 3A, may not be significantly higher due to about 0.6-fold higher than the 30μg dose group. In addition, the statistic method used to calculate the p value is required in Figure 3 legend, and fold difference in the text. 

In lines 298 – 300, compared to the saline-immunized macaques, the observation that UB-612 immunized macaque sera promoted stronger ADNKA activities against WA strain only shown in Figure 6C-D, rather than in Figure 6E.

Figures 5C, 6C, 6D and 6E are better combined as one Figure, and rest of Figures 5A-B and 6A-B are considered as supplemental figures.

4.    In the Discussion, it is imperative for authors to combine the results in previous study (A novel RBD-protein/peptide vaccine elicits broadly neutralizing antibodies and protects mice and macaques against SARS-CoV-2) to interpret current results. For instance, the T cell responses also contribute to protecting NHPs against SARS-CoV-2 replication in the respiratory tract.

In lines 345 – 346, for the message of “In the current study, we evaluated the activity of RBD-specific non-neutralizing antibodies”, the authors used either immunized sera or BAL in all activity or function assays, rather than using RBD-specific non-neutralizing antibodies, thus the results should be interpreted as the activity for both RBD-specific neutralizing and non-neutralizing antibodies. 

In lines 368 – 369,   “and we believe these high-avidity antibodies largely contributed to the breadth of neutralization against multiple variants of concern [17-20].” The authors did not test the breadth of neutralization against multiple variants of concern in this manuscript.

The lines 377 – 379, the authors only tested RBD-specific binding IgG but not S-specific binding IgG in this manuscript. In addition, the authors only tested Neutralizing antibody titers against SARS-CoV-2-WA but did not test the high breadth including against Omicron.

The lines 382 – 383, the sentence is not a scientific rationale due to using a word “likely”. In addition, regarding “The breadth of antibody response functionality after UB-612 immunization is important in protecting NHPs against SARS-CoV-2 replication in the respiratory tract.”, this sentence is always true and not meaningful. 

Minor suggestions:

In lines 71, using “induced” is more appropriate than using “induces” in the sentence of “Both preclinical and clinical studies demonstrated that UB-612 70 induces robust RBD- and S-specific antibodies ¼”.

In lines 257 – 258, Using “in the vaccine group but not in the saline group.” is clearer than “in the vaccine but not the saline groups.” 

In line 355, the sentence of “Other COVID-19 vaccines were also shown to induce ¼” is better altered to “Other COVID-19 vaccines also showed to induce ¼

The formats for the references #15 and #25 are not well done. 

Comments on the Quality of English Language

The quality of English language is good, but there are several occurrences such as some grammatical or meaningful confusion that require a careful proof prior to publication.

Author Response

In this manuscript, the authors report the characterization of the mucosal and Fc-mediated antibody responses elicited with RBD-protein/peptide vaccine UB-612 against SARS-CoV-2 in cynomolgus macaques. And in addition to the potent serum neutralizing and ACE2 binding inhibition antibodies, the mucosal antibodies and antibodies with Fc-mediated effector functions in antibody-dependent monocyte phagocytosis and antibody-dependent NK cell activation also contribute to the protection. Although the similar studies relevant to other types of vaccines have been reported, the results in this manuscript may prove that UB-612 is a promising COVID-19 vaccine that may provide a protection for the breath variants. The results overall are interesting.

The following suggestions are majorly intended to improve the information accuracy of the manuscript written and result interpretation.

  1. In the Abstract, the authors did not clarify the question or study goal that this study targeted to solve, in addition to lacking the method information. 

 The abstract has been amended to include the goal of the study and a short statement of the methods used to address the question.

  1. Lacking immunization method (inject route) information in the section of Materials and Methods for the immunization of animal cynomolgus macaques.

The route of injection has been introduced in M&M – intramuscular in the left deltoid muscle. If it was found unsuitable, then the choice was the right side. 

  1. In the section of Results.

In Figure 2A, by eye measuring, the RBD binding IgG titer (EC50) detected in UB-612 30μg dose group is slightly higher than that detected in UB-612 100μg dose group. In the text line 215, the RBD-specific antibody response (mean titer 1:47,544) detected in the UB-612 30μg dose group is a little bit lower than that detected in the UB-612 100μg dose group (mean titer 1:51,333).

This is a great catch; thank you! Indeed, the text included values that did not match the results presented in Figure 2A. I reviewed the most recent data files that were used to generate the illustrations, corrected the text values accordingly, and QC’ed beyond Figure 2 just in case, too.

In lines 237 – 239, the ACE-2 binding inhibition antibody assessments showed that the 100μg dose group (mean ID50 titers of 2,527.2), although it was statistically higher than the 30μg dose group (ID50 titers of 1,588.2 < 0.0001) in Figure 3A, may not be significantly higher due to about 0.6-fold higher than the 30μg dose group. In addition, the statistic method used to calculate the p value is required in Figure 3 legend, and fold difference in the text. 

Changed the text by removing “significant” wording, which in the context might have dual interpretation. Added the value of fold increase in ACE-2 binding titers between the low and high vaccine doses. The method of stat analysis was added in the figure 3 legend.

In lines 298 – 300, compared to the saline-immunized macaques, the observation that UB-612 immunized macaque sera promoted stronger ADNKA activities against WA strain only shown in Figure 6C-D, rather than in Figure 6E.

Figures 5C, 6C, 6D and 6E are better combined as one Figure, and rest of Figures 5A-B and 6A-B are considered as supplemental figures.

Amended the text by indicating no treatment effect in ADNKA for TNF-α positive CD56 NK cells. Combined the bar graphs 5C, 6C, 6D, and 6E into a new Figure A-D. Figures 5A-B and 6A-B were placed as supplemental Figures 1A-B and 2A-B, respectively.

  1. In the Discussion, it is imperative for authors to combine the results in previous study (A novel RBD-protein/peptide vaccine elicits broadly neutralizing antibodies and protects mice and macaques against SARS-CoV-2) to interpret current results. For instance, the T cell responses also contribute to protecting NHPs against SARS-CoV-2 replication in the respiratory tract.

I amended the conclusions in the Discussions by highlighting the importance of vaccine-induced T-cell immunity to protect against SARS-CoV-2 replication in NHP. 

In lines 345 – 346, for the message of “In the current study, we evaluated the activity of RBD-specific non-neutralizing antibodies”, the authors used either immunized sera or BAL in all activity or function assays, rather than using RBD-specific non-neutralizing antibodies, thus the results should be interpreted as the activity for both RBD-specific neutralizing and non-neutralizing antibodies. 

Deleted the word “non-neutralizing” to include all vaccine RBD-specific antibodies.

In lines 368 – 369,   “and we believe these high-avidity antibodies largely contributed to the breadth of neutralization against multiple variants of concern [17-20].” The authors did not test the breadth of neutralization against multiple variants of concern in this manuscript.

I agree. We did not test a direct link between high avidity antibodies and neutralization and its breadth. What we meant to say is that high avidity antibodies are likely to constitute a substantial fraction of neutralizing antibodies, which, in turn, are responsible for the breadth of activity against VoC tested in the previous study.

The lines 377 – 379, the authors only tested RBD-specific binding IgG but not S-specific binding IgG in this manuscript. In addition, the authors only tested Neutralizing antibody titers against SARS-CoV-2-WA but did not test the high breadth including against Omicron.

Correct, the conclusions were overreaching to include the results of the previous study. Amended accordingly to limit conclusions around the current findings.

The lines 382 – 383, the sentence is not a scientific rationale due to using a word “likely”. In addition, regarding “The breadth of antibody response functionality after UB-612 immunization is important in protecting NHPs against SARS-CoV-2 replication in the respiratory tract.”, this sentence is always true and not meaningful. 

Modified the summary to emphasize the importance of new findings in building evidence of UB-612 vaccine differentiation.  

Minor suggestions:

In lines 71, using “induced” is more appropriate than using “induces” in the sentence of “Both preclinical and clinical studies demonstrated that UB-612 70 induces robust RBD- and S-specific antibodies ¼”.

Corrected

In lines 257 – 258, Using “in the vaccine group but not in the saline group.” is clearer than “in the vaccine but not the saline groups.” 

Corrected

In line 355, the sentence of “Other COVID-19 vaccines were also shown to induce ¼” is better altered to “Other COVID-19 vaccines also showed to induce ¼”

Corrected

The formats for the references #15 and #25 are not well done. 

The reference #15 is erroneous; it leads to a news article summarizing findings from the reference #16. Reference #15 has been deleted, and the reference number is updated accordingly. Reference #25 is updated in line with ASC standards.

Reviewer 2 Report

Comments and Suggestions for Authors

This paper reports on the effects of the UB-612 vaccine on serum and mucosal reactivity, focusing on antibody-dependent monocyte phagocytosis (ADMP) and antibody-18 dependent NK cell activation (ADNKA) assays. RBD-specific antibodies, predominantly of the IgG1 and IgA subclasses, were also confirmed. Therefore, the results presented here are relevant to developing novel vaccine modalities. A major limitation of this study is the absence of SARS-CoV-2 variants in the assays, which will be a focus for future research. The safety profile in vivo should be carefully monitored, and a brief comment on the adverse events in this preclinical study would be helpful. Overall, the description of the paper is appropriately detailed, and the results of the ongoing phase III trial are awaited.

Author Response

This paper reports on the effects of the UB-612 vaccine on serum and mucosal reactivity, focusing on antibody-dependent monocyte phagocytosis (ADMP) and antibody-18 dependent NK cell activation (ADNKA) assays. RBD-specific antibodies, predominantly of the IgG1 and IgA subclasses, were also confirmed. Therefore, the results presented here are relevant to developing novel vaccine modalities. A major limitation of this study is the absence of SARS-CoV-2 variants in the assays, which will be a focus for future research. The safety profile in vivo should be carefully monitored, and a brief comment on the adverse events in this preclinical study would be helpful. Overall, the description of the paper is appropriately detailed, and the results of the ongoing phase III trial are awaited.

Thank you for your review and the comments. UB-612 has a unique composition of drug substances designed to provide a differentiated immune profile. The goal of the current study was to expand our understanding of humoral immune responses after immunization with UB-612. The vaccine has been explored for safety in several in vivo models, and some of the data has been previously described (former ref #20, currently #19). In addition, UB-612 underwent a complete repro-tox characterization to enable the IND/CTA Phase 3 study and an application for market authorization. The safety data in vivo was encouraging but massive to address in a scientific journal. Also, we have generated safety favorable data in over 3,000 subjects who received UB-612 during Phase 1 and 2 studies, with the results already published (ref #18 and 19, currently #17 and #18).
Regarding the variants, we published the NHP study results with various VoC tested in a virus neutralization assay (ref #20, currently #19). In preclinical studies, UB-612 consistently stimulated very broad VNT responses. This data is further supported by our clinical evidence from the past Phase 1 and 2 studies (ref# 18 and 19, currently 18 and 19). 

Reviewer 3 Report

Comments and Suggestions for Authors

The authors described the peptide vaccine induces humoral and cellular responses in macaques from their recent study (published in 2022) with more in depth analysis of the immune responses specifically focusing on non-neutralizing antibody responses. Their vaccine using protein and peptides along with adjuvants CpG1 and aluminum phosphate for formulation induced good anti-RBD IgG1 and a low level of IgA responses.

The authors need to describe their methods more clearly as it is confusing why they include EC50 for antibody titer instead of endpoint titers.

the titer of IgG1 (figure 2C) is about 5 times less than IgG (Figure 2A), is it an error in the way the titers are calculated? other isotype IgGs are negligible (Figure 2C).

The description of CD56+ cells for the expression of CD107a, IFNg and TNFa needs to be edited, its not activity but it is the percentage of the cells they are measuring by flow cytometry.

Comments on the Quality of English Language

minor editing is needed

Author Response

The authors described the peptide vaccine induces humoral and cellular responses in macaques from their recent study (published in 2022) with more in depth analysis of the immune responses specifically focusing on non-neutralizing antibody responses. Their vaccine using protein and peptides along with adjuvants CpG1 and aluminum phosphate for formulation induced good anti-RBD IgG1 and a low level of IgA responses.

The authors need to describe their methods more clearly as it is confusing why they include EC50 for antibody titer instead of endpoint titers.

Thank you for catching this mistake. The results were expressed differently for analysis, and the final version (based on the IgG endpoint) was not accurately reflected in the manuscript and Figure 2. The M&M results and figures were corrected to align with the ELISA method, which for the purpose of publication was chosen as an endpoint dilution: the maximum dilution with a reading over the cut-off, 2x SD over negative control.

the titer of IgG1 (figure 2C) is about 5 times less than IgG (Figure 2A), is it an error in the way the titers are calculated? other isotype IgGs are negligible (Figure 2C).

This is accurate; the absolute titers of IgG and Ig1 in two separate experiments differed by a factor of 5. Without having an internal control bridging two separate studies, it is very challenging to interpret this difference other than by the variability between two independent test systems.

The description of CD56+ cells for the expression of CD107a, IFNg and TNFa needs to be edited, its not activity but it is the percentage of the cells they are measuring by flow cytometry.

Addressed in the M&M and the Results sections: ADNKA test results are presented as the proportion of activated CD56+ cells with the corresponding markers.

Round 2

Reviewer 1 Report

Comments and Suggestions for Authors

The results are interesting, and the quality of the manuscript is high.